# Epigenetic memory independent of symmetric histone inheritance

**Daniel S Saxton, Jasper Rine***

Department of Molecular and Cell Biology, University of California, Berkeley, Berkeley, United States

**Abstract** Heterochromatic gene silencing is an important form of gene regulation that usually requires specific histone modifications. A popular model posits that inheritance of modified histones, especially in the form of H3-H4 tetramers, underlies inheritance of heterochromatin. Because H3-H4 tetramers are randomly distributed between daughter chromatids during DNA replication, rare occurrences of asymmetric tetramer inheritance within a heterochromatic domain would have the potential to destabilize heterochromatin. This model makes a prediction that shorter heterochromatic domains would experience unbalanced tetramer inheritance more frequently, and thereby be less stable. In contrast to this prediction, we found that shortening a heterochromatic domain in *Saccharomyces* had no impact on the strength of silencing nor its heritability. Additionally, we found that replisome mutations that disrupt inheritance of H3-H4 tetramers had only minor effects on heterochromatin stability. These findings suggest that histones carry little or no memory of the heterochromatin state through DNA replication.
DOI: https://doi.org/10.7554/eLife.51421.001

## Introduction

A central question in biology is how cells with identical genotypes can exhibit different, heritable phenotypes. By definition, these phenotypes are determined by information that is epigenetic, or 'above the genome.' Just as genetic inheritance requires faithful replication of DNA, epigenetic inheritance requires replication of information that is transmitted to both daughter cells during division. Faithful transmission of epigenetic information is crucial for multiple heterochromatin-based processes such as X-chromosome inactivation in mammals and cold-induced gene silencing in *Arabidopsis*. In these cases and others, the epigenetic inheritance of heterochromatin indicates that some components of heterochromatin behave as heritable units. Surprisingly, the identity of this epigenetic information remains unclear and heavily debated.

The histone subunits of nucleosomes, especially histones H3 and H4, are marked by a variety of covalent modifications that are integral to heterochromatin function. During DNA replication, nucleosomes are partially disrupted and marked parental H3-H4 tetramers are locally inherited to daughter chromatids. As these tetramers are inherited, they are reassembled into nucleosomes that are interspersed with nucleosomes containing newly synthesized H3-H4 tetramers (*Prior et al., 1980*; *Jackson, 1988*; *Schlissel and Rine, 2019*). One model for epigenetic inheritance posits that marked parental histones inherited through DNA replication recruit histone modifiers to deposit similar marks on new adjacent nucleosomes, thereby reestablishing the previous local landscape of histone modifications (*Hecht et al., 1995*; *Hoppe et al., 2002*; *Gaydos et al., 2014*). In support of this model, the H3K27 methyltransferase PRC2 binds preferentially to H3K27me3 in vitro (*Hansen et al., 2008*) and some other modifying enzymes show a similar ability to bind their histone modifications (*Zhang et al., 2008*; *Hecht et al., 1995*; *Imai et al., 2000*). If this model is correct, modified H3-H4 tetramers would constitute heritable units that drive epigenetic memory of chromatin states.

**\*For correspondence:**
jrine@berkeley.edu

**Competing interests:** The authors declare that no competing interests exist.

**eLife digest** A crucial process in life is the ability of cells to pass on useful information to their descendants. Some of this information is encoded within molecules of DNA, including genes that contain specific coded instructions. Another layer of information helps to specify whether individual genes are switched on or off, which means cells with the same genes can perform different tasks. However, it remains unclear exactly how cells pass on this additional layer of "epigenetic" information.

Inside human, yeast and other eukaryotic cells, DNA is wrapped around scaffold proteins known as histones. Cells modify histones by adding chemical tags to them, and histones within the same gene often have specific patterns of chemical tags. One popular hypothesis is that these marked histones constitute epigenetic information that may be passed on when DNA replicates before a cell divides to make two daughter cells. This model predicts that the marked histones need to be divided equally between the two sets of DNA to allow the epigenetic information to be faithfully passed on to both daughter cells.

To test this prediction, Saxton and Rine studied a gene called *HMR* that is involved in mating in yeast. This gene is constantly silenced (in other words, not actively providing instructions to the cell) and contains histones with very specific patterns of chemical tags. For the experiments, Saxton and Rine made a series of mutations in the yeast that increased how often these marked histones were divided unequally when the yeast cells replicated their DNA. Unexpectedly, these mutations had little impact on the ability of the cells to pass on the silenced state of *HMR* to their offspring. These findings argue against the classic model that marked histones carry epigenetic information.

DOI: https://doi.org/10.7554/eLife.51421.002

Studies have come to different conclusions regarding whether histones can carry epigenetic memory. In *S. pombe,* localized methylation of H3K9 can silence a reporter gene, and this silenced state is heritable in the presence of the H3K9 methyltransferase Clr4p as long as the demethylase Epe1p is absent (*Audergon et al., 2015*; *Ragunathan et al., 2015*). These studies suggest that histone modifications can facilitate epigenetic inheritance, and caution that such a mechanism is normally obscured by H3K9 demethylation activity. Conversely, induced removal of silencer elements from silenced chromatin in *S. cerevisiae* causes almost all cells to lose silencing of adjacent genes after just one round of DNA replication (*Holmes and Broach, 1996*). Similar results are found when silencers are removed from *Drosophila* chromatin silenced by the Polycomb complex (*Laprell et al., 2017*). These silencer-removal experiments suggest that modified histones are not sufficient to propagate the silenced chromatin state through DNA replication.

The model in which histones carry epigenetic memory makes a testable prediction: since parental H3-H4 tetramers have long been thought to be randomly partitioned between daughter chromatids (*Sogo et al., 1986*; *Cusick et al., 1984*), rare events could occur in which most or all marked parental H3-H4 tetramers within a domain segregate asymmetrically to one daughter chromatid, causing the other to inherit primarily newly synthesized histones. A chromatin domain with an insufficient number of marked parental tetramers would be expected to experience a loss-of-chromatin-state event. In this view, a smaller chromatin domain would correspond to fewer marked nucleosomes and yield more frequent events in which parental H3-H4 tetramers segregate asymmetrically and the chromatin state is lost. This potential use of domain size for protection against epimutation is widely conjectured (*Dodd et al., 2007*; *Kaufman and Rando, 2010*; *Moazed, 2011*; *Ramachandran and Henikoff, 2015*), and may explain why chromatin domains subject to stable epigenetic inheritance are often many kilobases long. For example, chromatin domains silenced by Polycomb Responsive Elements (PREs) in *Drosophila* usually extend beyond 10 kb (*Schwartz et al., 2006*). In contrast, one study in *A. thaliana* found that a chromatin domain containing only three H3K27me3-marked nucleosomes is inherited more frequently than would be predicted if random segregation of tetramers caused loss events (*Yang et al., 2017*). However, no study to our knowledge has systematically tested this prediction.

To test directly whether inheritance of a chromatin state is affected by chromatin domain size, we focused on the heterochromatin domains at the *HMR* and *HML* loci in *S. cerevisiae*. These loci

contain copies of mating-type genes that are silenced by the activity of Sir proteins. Specifically, the *E* and *I* silencers flanking *HMR* and *HML* are occupied by the DNA-binding proteins Rap1, Abf1, and ORC, that collectively recruit Sir proteins; Sir1 is present only at silencers, whereas Sir2/3/4 complexes bind to silencers and spread across the locus in a process that requires deacetylation of H4K16 (*Rusché et al., 2002*; *Thurtle and Rine, 2014*). Notably, DNA methylation and RNA interference do not exist in *S. cerevisiae*.

Under normal conditions, *HMR* and *HML* are constitutively silenced. Rare and transient loss-of-silencing events can be measured by a sensitive assay that uses the *cre* recombinase under control of the *HMLα2* promoter to convert transient transcriptional events into permanent, heritable changes in fluorescence phenotypes (*Dodson and Rine, 2015*). In contrast, deletion of *SIR1* causes genetically identical cells to be in either of two states at *HMR* and *HML*: either fully silenced or fully expressed (*Pillus and Rine, 1989*; *Xu et al., 2006*; *Dodson and Rine, 2015*). These different transcriptional states are mitotically heritable and cells switch between states at a low frequency. This study addresses three questions regarding the inheritance of heterochromatin in *Saccharomyces*: 1) Does the size of a silenced domain determine the fidelity of inheritance? 2) Does removal of Sir1, a protein that facilitates recruitment of silencing machinery to silencers, uncover an effect of chromatin domain size on heritability of transcriptional states? 3) Do replisome components that facilitate symmetric inheritance of parental H3-H4 tetramers also promote inheritance of transcriptional states?

## Results

Local inheritance of histones and their locus-specific modifications are thought to facilitate inheritance of chromatin states. According to this view, if parental H3-H4 tetramers were randomly partitioned between the two daughter chromatids during replication, one would expect a chromatin state to be lost if, by chance, one of the daughter chromatids failed to receive enough parental H3-H4 tetramers to support the propagation of that state. By this model, the number of nucleosomes in the chromatin domain would influence the fidelity of chromatin-state inheritance.

### Nucleosome number did not determine the rate of silencing loss

To test if nucleosome number affected the stable inheritance of a chromatin state, we used the Cre-Reported Altered States of Heterochromatin (CRASH) assay (*Dodson and Rine, 2015*) (*Figure 1A*). In this assay, *cre* replaces the *α2* coding sequence in *HMRα*, and a *lox* cassette containing fluorescent reporters separated by *loxP* sites is located on a separate chromosome. Though *HMRα* is transcriptionally repressed, rare loss-of-silencing events cause transient expression of *cre*. These events lead to excision of *RFP* from the lox cassette, and a switch from RFP to GFP expression. Because this change is heritable, loss-of-silencing events during colony growth lead to formation of sectors of cells expressing GFP, appearing green on an otherwise red background. The number of sectors in a colony reflects the frequency at which *HMRα* transiently loses silencing: more sectors indicate less stable silencing.

*HMRα::cre* contained fourteen well-positioned nucleosomes between the *E* and *I* silencers (*Figure 1—figure supplement 1*). To change nucleosome number within the locus, we deleted DNA corresponding to different sets of nucleosomes (*Figure 1B*). Notably, removing DNA corresponding to different combinations of well-positioned nucleosomes allowed us to discern whether any effects on silencing stability were due to nucleosome number or to removal of specific DNA sequences. These deletions did not affect the local positions of the remaining nucleosomes as measured by MNase-Seq (*Figure 1—figure supplement 1*).

At the limit of models by which nucleosomes transmit memory of transcriptional states, inheritance of a single parental H3-H4 tetramer to a daughter chromatid would be sufficient to template the silenced state. The expected loss-of-silencing rate would thereby reflect the frequency at which a chromatid inherits no marked parental H3-H4 tetramers due to random segregation of these tetramers during replication. For example, considering a hypothetical chromatin domain that has three nucleosomes, one would expect that a given daughter chromatid would have a one-in-eight chance of inheriting no parental tetramers during replication. Therefore, one in eight daughter cells would be expected to lose silencing. This rate would increase exponentially with shorter chromatin domains as the probability of inheriting at least one parental tetramer decreases (*Figure 1C*). Additionally, if

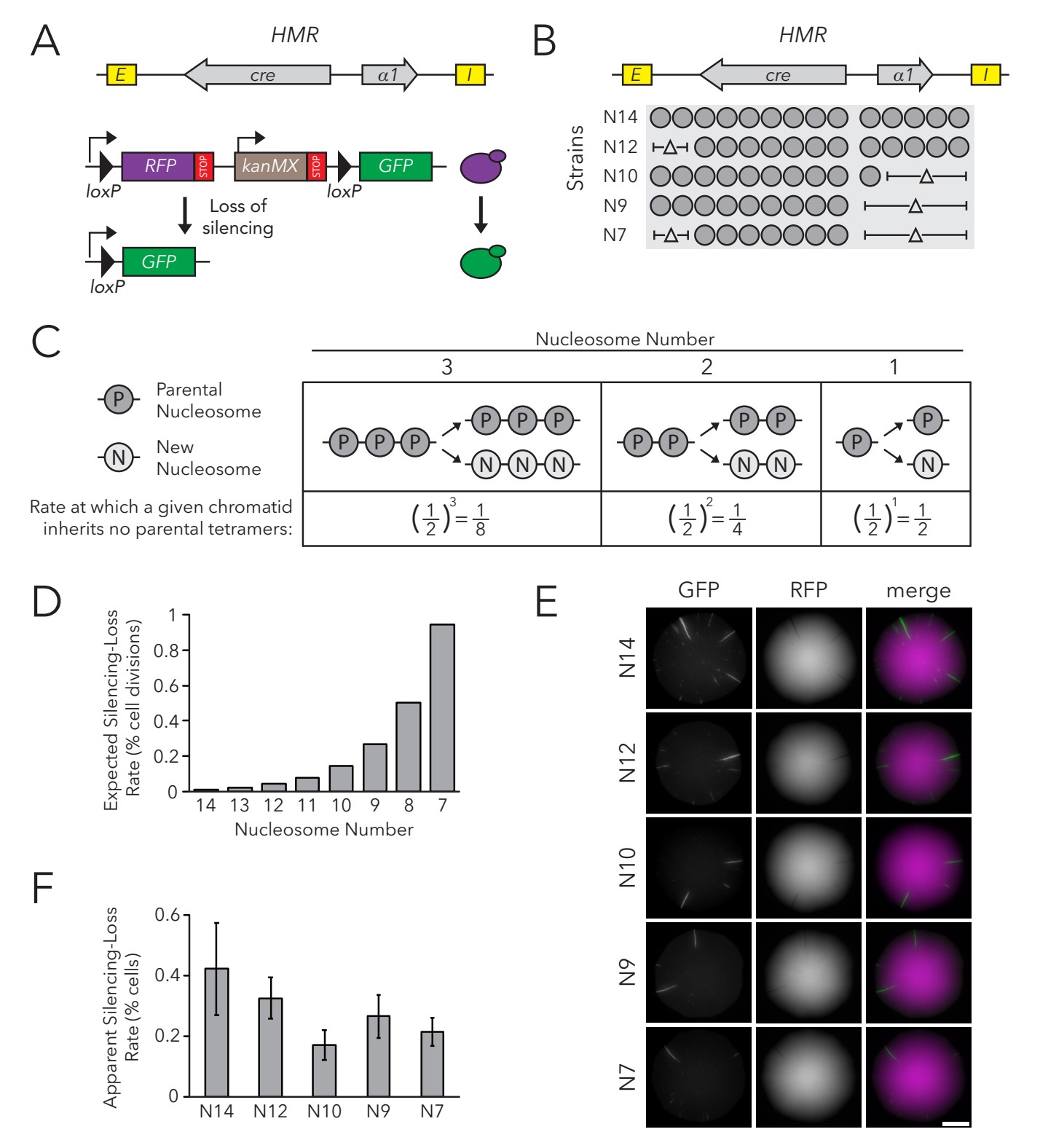

**Figure 1.** Chromatin Domain Size Did Not Affect Silencing-Loss Rates. (**A**) Schematic of the Cre-Reported Altered States of Heterochromatin (CRASH) assay (*Dodson and Rine, 2015*). *HMRα::cre* contains the *E* and *I* silencers, the *α1* gene, and a *cre* transgene. Transient loss of silencing at *HMRα::cre* causes Cre-mediated recombination of *loxP* sites in a *RFP-GFP* cassette. This process creates a permanent, heritable switch from RFP to GFP expression. (**B**) Diagram of nucleosomes in *HMRα::cre*. Fourteen nucleosomes were present in full-length *HMRα::cre*, which we term Strain N14

*Figure 1 continued on next page*

*Figure 1 continued*

(JRY11471). Combinations of nucleosomal DNA were deleted to change the size of *HMRα::cre*; the smallest allele contained seven nucleosomes (Strain N7) (JRY11540). Nucleosome positions were determined by MNase-Seq as shown in *Figure 1—figure supplement 1*. (C) Schematic of how random segregation of parental H3-H4 tetramers to daughter chromatids could cause silencing loss. Under the model that inheritance of a single marked H3-H4 tetramer to a daughter chromatid would be sufficient to propagate the silenced state, the chance that a daughter chromatid inherits no parental tetramers and loses the silenced state would be 0.5^(the number of nucleosomes in the chromatin domain). Parental nucleosomes contain inherited H3-H4 tetramers, whereas new nucleosomes contain newly synthesized H3-H4 tetramers. Hypothetical chromatin domains of different sizes are provided for comparison. (D) Expected loss-of-silencing rates for different sizes of *HMRα::cre*. (E) Representative CRASH colonies for Strains N14 through N7. Because loss of silencing leads to a heritable switch from RFP to GFP expression, progeny of a cell that loses silencing will form a GFP sector; the frequency of sectors in a colony represents the frequency at which that strain loses silencing. Scale bar, 2 mm. (F) Quantification of apparent silencing-loss rates, as described in Materials and methods. Data are means ± SD (n = 6 independent cultures). ANOVA and Tukey tests were used to test statistical significance. Only strains N10 and N7 were significantly different (p<0.05) than N14. Data are presented as a scatter plot in *Figure 1— figure supplement 2*.

DOI: https://doi.org/10.7554/eLife.51421.003

The following figure supplements are available for figure 1:

**Figure supplement 1.** Nucleosome set deletions did not affect positions of remaining nucleosomes.
DOI: https://doi.org/10.7554/eLife.51421.004
**Figure supplement 2.** Chromatin domain-size of *HMRα::cre* had minimal effects on silencing.
DOI: https://doi.org/10.7554/eLife.51421.005
**Figure supplement 3.** *HMLα::cre* contained 22 nucleosomes.
DOI: https://doi.org/10.7554/eLife.51421.006
**Figure supplement 4.** Chromatin domain size of *HMLα::cre* had minimal effects on silencing-loss rates.
DOI: https://doi.org/10.7554/eLife.51421.007
**Figure supplement 5.** Chromatin domain size of *HMLα::cre* had minimal effects on silencing-loss rates measured by flow cytometry.
DOI: https://doi.org/10.7554/eLife.51421.008

inheritance of two or more parental H3-H4 tetramers was necessary to template the silenced state, the expected loss-of-silencing rate would be even higher.

The silencing-loss rate predicted by random segregation of H3-H4 tetramers would be approximately 0.006% of cell divisions for full-length *HMRα::cre* (Strain N14) (*Figure 1D*). Previous studies demonstrate that this strain loses silencing in approximately 0.1% of cell divisions (*Dodson and Rine, 2015*). This difference between expected and observed values could be explained by the existence of other processes besides histone inheritance that potentially destabilize silencing and thereby contribute to the overall silencing-loss rate. In contrast to the full size *HMRα::cre*, the smallest allele of *HMRα::cre* (Strain N7) would be expected to lose silencing in approximately 1% of cell divisions (*Figure 1D*). Therefore, if this model were correct, we would expect to see increased sectoring rates in strains with shorter alleles of *HMRα::cre*. Surprisingly, decreasing nucleosome number at *HMRα::cre* led to a slight decrease in silencing loss as measured by sector frequency (*Figure 1E*).

To provide an independent measurement of the silencing-loss rate, we also measured fluorescence profiles of single cells. Cells that have recently lost silencing of *cre* at *HMRα* contain both RFP and GFP due to GFP expression and the persistence of RFP prior to its degradation and dilution. Using flow cytometry to measure the frequency of cells that contain both RFP and GFP, we confirmed that nucleosome number did not strongly affect silencing-loss rates, and that reduction of nucleosomes might have a slight stabilizing effect on silencing (*Figure 1F*, *Figure 1—figure supplement 2*). Thus, the size of *HMRα::cre* did not dramatically influence inheritance of the silenced state, in contrast to the expectation from models in which H3-H4 tetramers carry memory of chromatin states through cell divisions. Additionally, we found that changing nucleosome number at *HMLα::cre* led to only a small increase in silencing loss, and that these effects were not due strictly to domain size (*Figure 1—figure supplements 3–5*). Since studies at *HMLα* are potentially complicated by its proximity to a telomere, which is also bound by Sir proteins, further studies were performed only at *HMRα*.

## Nucleosome number did not affect transmission of epigenetic states in *sir1Δ*

The silencers flanking *HMRα* are bound by three different proteins that collaborate to recruit Sir proteins (*Rusche et al., 2003*). One possibility for the apparent insensitivity of silencing inheritance to

nucleosome number was that the constant recruitment of Sir proteins to these sites was efficient enough to mask a contribution of histone inheritance to inheritance of chromatin states. In this scenario, silencers would be capable of recruiting enough Sir proteins to keep the locus silenced during DNA replication, regardless of histone segregation patterns. Sir1 binds to silencers, and deletion of *SIR1* partially disrupts silencer activity, as measured by defects in silencing establishment and silencing heritability (*Pillus and Rine, 1989*; *Dodson and Rine, 2015*). We therefore tested if parental H3-H4 tetramer inheritance contributed to transmission of the silenced state when silencer-based recruitment of Sir proteins was impaired by the *sir1Δ* mutation.

Within individual cells in a population of *sir1Δ* cells, *HMR* is either transcriptionally silenced or fully expressed. These different states are mitotically heritable: a cell in one state usually gives rise to more cells of that state. To observe this epigenetic phenomenon, we placed the *GFP* coding sequence into *HMRα*, such that it was expressed under control of the *α2* promoter. Silencing was monitored by GFP expression at the single-cell level using fluorescence microscopy and flow cytometry. In comparison to control strains in which *HMRα* was fully silenced (*SIR+*) or expressed (*sir4Δ*), *HMRα* was silenced in roughly 99% of *sir1Δ* cells and was expressed in the remaining cells (*Figure 2—figure supplements 1* and *2*). We also observed different epigenetic states for *HMLα::RFP*. We used live-cell imaging to monitor divisions of *sir1Δ* cells to identify cells in which silencing of *HMR* was lost, and other cases in which it was gained (*Figure 2A*, *Video 1*). Thus *HMRα::GFP* could be used to measure the efficiency of epigenetic inheritance in *sir1Δ*, similarly to previous studies (*Xu et al., 2006*). For simplicity, we named measurements of epigenetic inheritance in *sir1Δ* as the FLuorescent Analysis of Metastable Expression (FLAME) assay, which is commonly implemented by live cell microscopy but is also adapted to flow cytometry as noted in individual experiments.

To test the prediction that chromatin domain size affects silencing heritability with the FLAME assay, we removed DNA corresponding to sets of nucleosomes in the *HMRα::GFP* locus (*Figure 2B*, *Figure 2—figure supplements 3* and *4*). As before, models in which nucleosomes were carriers of epigenetic memory predicted that shorter chromatin domains would have a higher rate of silencing loss (*Figure 2C*). Using time-lapse fluorescence microscopy to monitor transcriptional states in individual cells and their descendants as they divided, we found that nucleosome number did not affect the frequency of silencing loss (*Figure 2D*). Because the expressed state is also heritable, with occasional switches to the silenced state, we also asked if the heritability of the expressed state was influenced by the number of nucleosomes in the locus. The frequency of silencing establishment was similar between strains with different numbers of nucleosomes at *HMRα::GFP* (*Figure 2E*). Therefore, even in a background with defective silencer activity, chromatin-domain size did not strongly influence silencing dynamics. These findings argued against models in which parental H3-H4 tetramers and their modifications are required for the epigenetic inheritance of gene expression states in *Saccharomyces*.

## Replisome defects affected epigenetic inheritance

An orthogonal approach to test the role of histones in carrying epigenetic memory would be to consistently bias parental H3-H4 tetramer inheritance to one daughter chromatid, leaving the other daughter chromatid with fewer parental H3-H4 tetramers. Recent reports demonstrate conserved roles of two replisome components, Dpb3 and Mcm2, in producing a more symmetric distribution of parental H3-H4 tetramers between the leading and lagging strands. Specifically, *dpb3Δ* causes biased parental H3-H4 tetramer inheritance to the lagging strand (*Yu et al., 2018*) and a set of point mutations in *MCM2* (*mcm2-3A*) causes biased parental H3-H4 tetramer inheritance to the leading strand (*Petryk et al., 2018*; *Gan et al., 2018*). A complementary study found that local histone H4 inheritance in a small chromatin domain was moderately reduced in both the *dpb3Δ* and *mcm2-3A* single mutants, and severely reduced in the *dpb3Δ mcm2-3A* double mutant (*Schlissel and Rine, 2019*). Together, these studies demonstrate that Dpb3 and Mcm2 are necessary for efficient inheritance of parental H3-H4 tetramers to both daughter chromatids during DNA replication.

If parental H3-H4 tetramer inheritance contributes to transmission of chromatin states, we would predict more loss-of-silencing events in strains with defects in tetramer inheritance. To test this idea, we measured silencing loss in replisome mutants using the CRASH assay (*Figure 3A*). The *dpb3Δ* and *mcm2-3A* single mutants exhibited higher silencing-loss rates, consistent with previous studies done at *HML* (*Yu et al., 2018*; *Gan et al., 2018*), and the *dpb3Δ mcm2-3A* double mutant lost silencing more frequently than either single mutant. Similar results were obtained by using flow

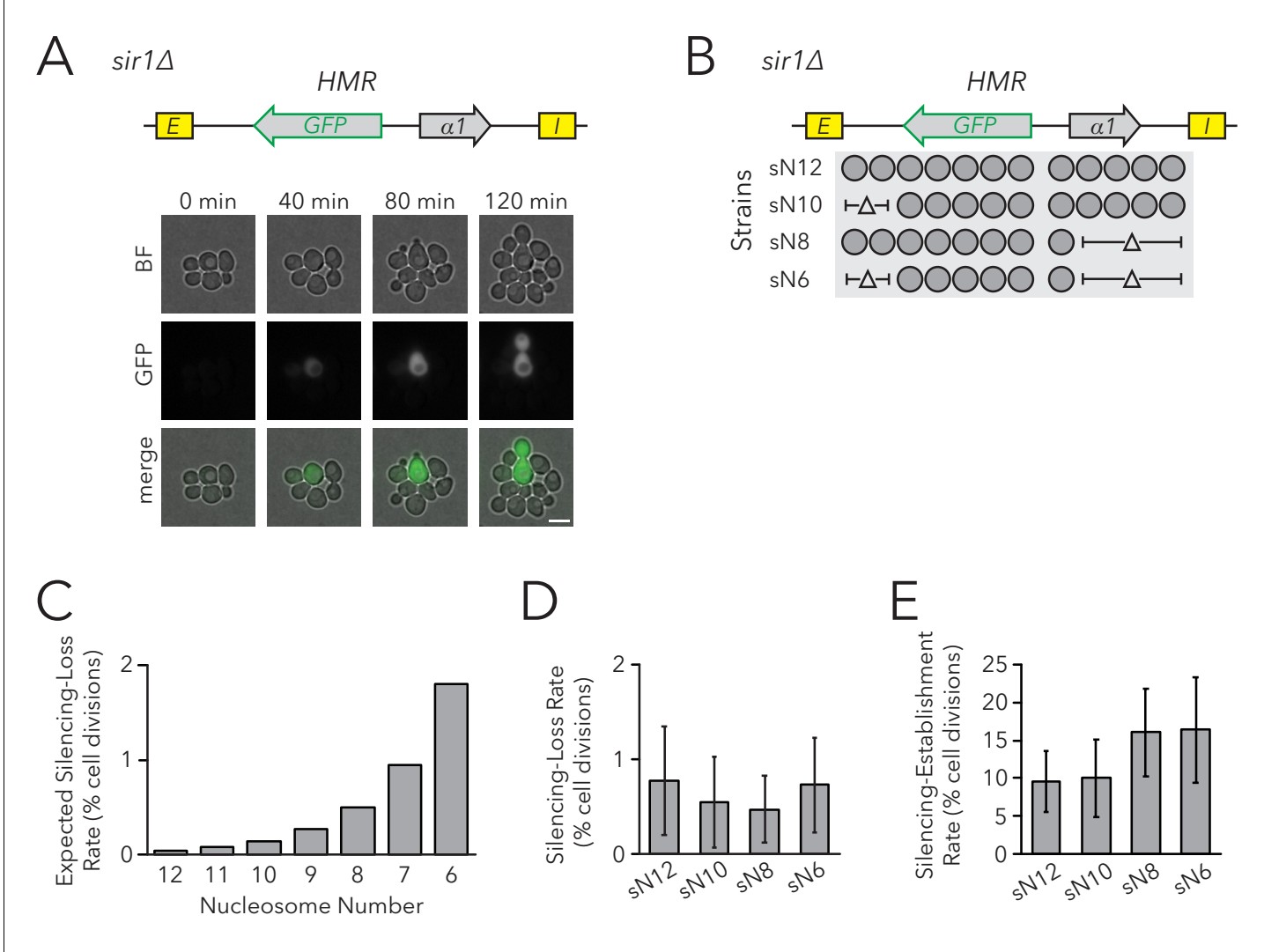

**Figure 2.** Chromatin Domain Size Did Not Affect Silencing-Loss Rates in *sir1Δ*. (**A**) Diagram of the FLuorescent Analysis of Metastable Expression (FLAME) assay. In a *sir1Δ* background, *GFP* replaced the *α2* gene so that transcriptional activity of *HMRα::GFP* could be monitored at the single-cell level (JRY11478). Loss-of-silencing events were observed in dividing cells by using time-lapse microscopy. Scale bar, 5 μm. Establishment-of-silencing events were also observed (see *Video 1*). Silencing defects in different *sir* mutants are shown by microscopy in *Figure 2—figure supplement 1* and by flow cytometry in *Figure 2—figure supplement 2*. (**B**) Diagram of nucleosomes in *HMRα::GFP* as defined by MNase-Seq (*Figure 2—figure supplement 3*). Twelve nucleosomes were present in full-length *HMRα::GFP* (Strain sN12) (JRY11478). Combinations of nucleosomal DNA were deleted to change the size of the *HMRα::GFP* locus; the smallest allele contained six nucleosomes (Strain sN6) (JRY11547). (**C**) Expected loss-of-silencing rate from random segregation of H3-H4 tetramers to daughter chromatids. See the legend of *Figure 1* for a description of how these expected rates were calculated. (**D**) Observed loss-of-silencing rates using the FLAME assay. Cell divisions were monitored by time-lapse microscopy (n > 900 cell divisions per genotype). Silencing-loss rates were not significantly different (Yates chi-square test, p>0.05 for all pairwise comparisons). (**E**) Observed establishment-of-silencing rates using the FLAME assay (n > 110 cell divisions per genotype). Silencing establishment rates were not significantly different (Yates chi-square test, p>0.05 for all pairwise comparisons). These strains showed similar frequencies of silenced and expressed cells as measured by flow cytometry in *Figure 2—figure supplement 4*. Error bars represent 95% confidence intervals.

DOI: https://doi.org/10.7554/eLife.51421.009

The following figure supplements are available for figure 2:

**Figure supplement 1.** *sir1Δ* cells exhibited metastability at *HMRα::GFP*.

DOI: https://doi.org/10.7554/eLife.51421.010

**Figure supplement 2.** *sir1Δ* cells exhibited metastability at *HMLα::RFP* and *HMRα::GFP*.

DOI: https://doi.org/10.7554/eLife.51421.011

**Figure supplement 3.** *HMRα::GFP* contained 12 nucleosomes.

DOI: https://doi.org/10.7554/eLife.51421.012

*Figure 2 continued on next page*

*Figure 2 continued*

**Figure supplement 4.** Chromatin domain size of *HMRα::GFP* in *sir1Δ* did not affect the frequencies of different epigenetic states but did affect GFP expression levels.

DOI: https://doi.org/10.7554/eLife.51421.013

cytometry to measure silencing-loss rates (*Figure 3B*). These data were consistent with a model in which inheritance of parental H3-H4 tetramers could contribute to inheritance of the silenced state at *HMR*. However, the data were also compatible with the possibility that heterochromatin assembled in such mutants was simply unstable for reasons independent of defects in its inheritance. Additionally, since previous studies did not specifically test the effects of Dpb3 and Mcm2 on histone inheritance within heterochromatin, any interpretations of silencing defects operated under the assumption that these replisome components act similarly between heterochromatin and euchromatin.

It is possible that parental H3-H4 tetramer inheritance affects both transient loss-of-silencing events, as detected by the CRASH assay, and heritability of epigenetic states. Testing this possibility was important because the currently unidentified epigenetic information that determines expression states in *sir1Δ* is transmitted locally at *HML* and *HMR*, respectively, rather than being transmitted in *trans* from processes elsewhere in the cell (*Xu et al., 2006*). If parental H3-H4 tetramers were the crucial local factors that transmitted this information, we would predict that disrupted tetramer inheritance would cause more loss-of-silencing events in *sir1Δ*. To test this possibility, we generated replisome mutant strains in combination with *sir1Δ* and evaluated the inheritance of transcriptional states using two different FLAME assay measurements: Fluorescence-Activated Cell Sorting (FACS) and live-cell microscopy.

Populations of *dpb3Δ*, *mcm2-3A*, and *dpb3Δ mcm2-3A* mutants all showed a mix of cells that were silenced or expressed at *HMRα::GFP*; all three mutant strains also showed a higher frequency of expressed cells than wild type (*Figure 4—figure supplement 1*, *Table 1*). Because silencing-loss rates and silencing-establishment rates both affect the frequency of cells in which *HMR* is silenced or expressed, one or both of these rates were presumably different in replisome mutants. To measure these rates, we used FACS to sort cells from each strain into two separate populations of *HMR*-silenced and *HMR*-expressed cells, and used flow cytometry to monitor the rates at which these initial sorted populations relaxed back to a mixed population of silenced and expressed cells (*Figure 4A*). These relaxation rates, and the frequency of silenced cells at equilibrium, were products of competing silencing-loss and silencing-establishment rates. By using these relaxation rates to calculate silencing-loss rates (*Figure 4B*, *Figure 4—figure supplement 2*), we observed that *dpb3Δ* and *mcm2-3A* had higher loss-of-silencing rates than wild type (*Figure 4C*). The *dpb3Δ mcm2-3A* double mutant had a higher loss rate than the single mutants. Similar loss trends were observed using time-lapse fluorescence microscopy (*Figure 4D*), albeit with overall higher loss rates than those seen with FACS. Together, these data suggested that faithful inheritance of parental H3-H4 tetramers helped transmit the silenced state of *HMR*. However, we also noted that the vast majority of silenced cells still faithfully transmitted the silenced state in the replisome mutant backgrounds.

We also asked if replisome mutants had differences in the frequency of silencing-establishment events. Curiously, any strain containing

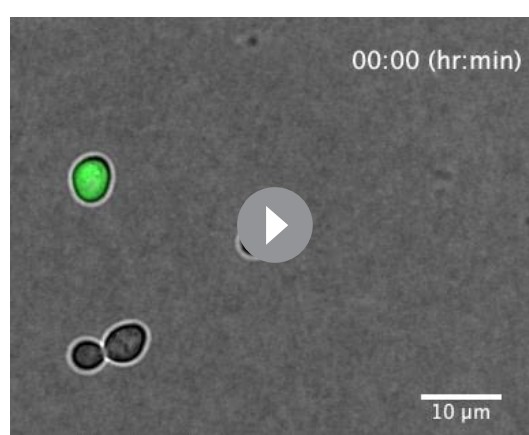

**Video 1.** Time-lapse video of inheritance of epigenetic states in the FLAME assay. *HMRα::GFP sir1Δ* (JRY11478) cells were grown to log-phase in liquid medium and subsequently imaged by time-lapse microscopy. A loss-of-silencing event is visible near the center of the field of view at 4 hr, and an establishment-of-silencing event is visible near the upper-left corner at 5 hr.

DOI: https://doi.org/10.7554/eLife.51421.014

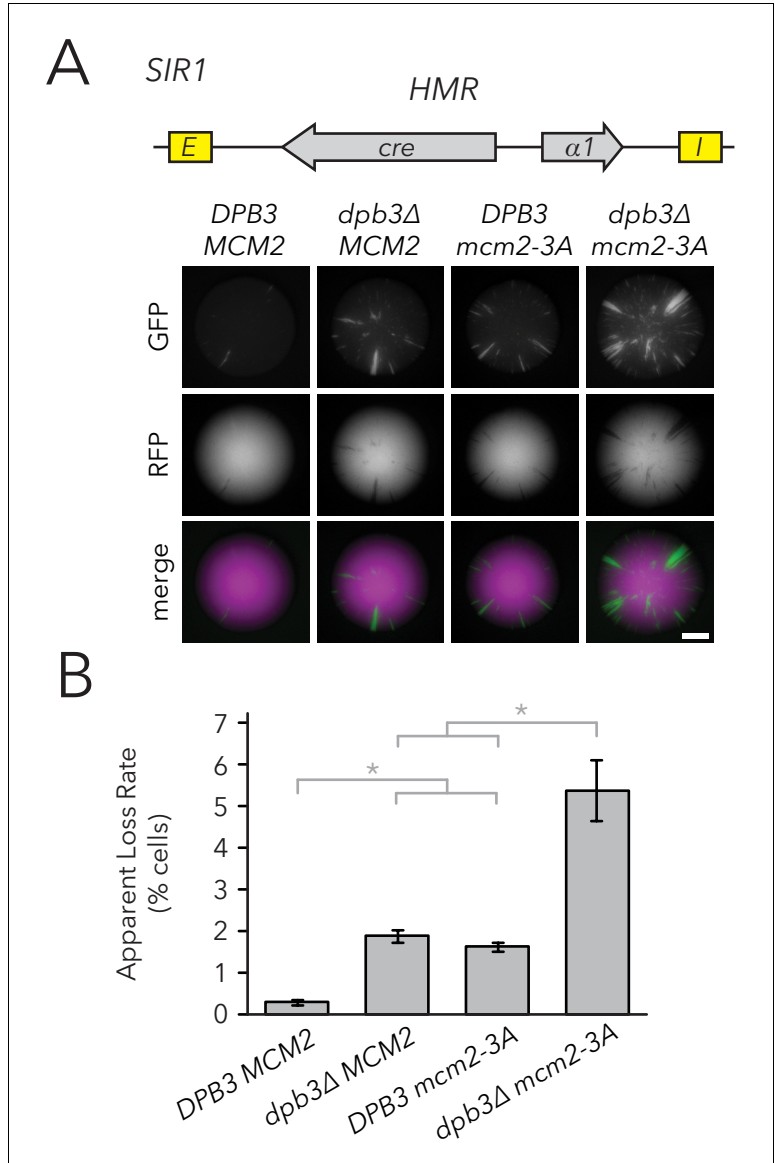

**Figure 3.** Replisome mutants exhibited higher silencing-loss rates in the CRASH assay. (**A**) Representative CRASH colonies for *DPB3 MCM2* (JRY11471), *dpb3Δ MCM2* (JRY11562), *DPB3 mcm2-3A* (JRY11591), and *dpb3Δ mcm2-3A* (JRY11592). Scale bar, 1 mm. (**B**) Quantification of apparent silencing-loss rates of strains in (**A**), as described in Materials and methods. Data are means ± SD (n = 6 independent cultures). ANOVA and Tukey tests were used to test statistical significance. *DPB3 MCM2* was significantly different than *dpb3Δ MCM2* and *DPB3 mcm2-3A* (p<0.05 each), and *dpb3Δ mcm2-3A* was significantly different than *dpb3Δ MCM2* and *DPB3 mcm2-3A* (p<0.05 each).
DOI: https://doi.org/10.7554/eLife.51421.015

*dpb3Δ* had an increased establishment rate, whereas *mcm2-3A* had minimal, if any, effects on establishment rate (*Figure 4E–G*). Additionally, any strain containing *dpb3Δ* showed elevated levels of *HMRα::GFP* expression in unsilenced cells, as measured by flow cytometry (*Figure 4—figure supplement 3*). Because *dpb3Δ* cells more readily established silencing, we inferred that the expressed state was less efficiently inherited. Therefore, Dpb3 contributed to the inheritance of the expressed state of *HMR* as well as to the silenced state.

**Table 1.** Comparison of epigenetic switching rates and proportion of silenced cells at equilibrium.

Data for *DPB3 MCM2* (JRY11471), *dpb3Δ MCM2* (JRY11550), *DPB3 mcm2-3A* (JRY11589), and *dpb3Δ mcm2-3A* (JRY11590) in the FLAME assay was extracted from *Figure 4*. The percentages of Silenced (S) and Expressed (E) cells at equilibrium were determined from *Figure 4B*. Silencing-loss rates ($k_{on}$, $gen^{-1}$) correspond to data from *Figure 4C* and silencing-establishment rates ($k_{off}$, $gen^{-1}$) correspond to data from *Figure 4F*. If $k_{on}$ and $k_{off}$ rates accurately predict the percentages of silenced and expressed cells at equilibrium, then E/S should be similar to $k_{on}/k_{off}$.

| Genotype | Silenced (S) | Expressed (E) | S → E ($k_{on}$, $gen^{-1}$) | E → S ($k_{off}$, $gen^{-1}$) | E/S | $k_{on}/k_{off}$ |
|---|---|---|---|---|---|---|
| *DPB3 MCM2* | 99 | 1 | 0.003 | 0.11 | 0.01 | 0.031 |
| *dpb3Δ MCM2* | 96 | 4 | 0.014 | 0.78 | 0.042 | 0.019 |
| *DPB3 mcm2-3A* | 91 | 9 | 0.018 | 0.17 | 0.094 | 0.107 |
| *dpb3Δ mcm2-3A* | 90 | 10 | 0.033 | 0.5 | 0.115 | 0.066 |

DOI: https://doi.org/10.7554/eLife.51421.020

## Variations in nucleosome number in replisome mutant backgrounds

Though the rate of silencing loss increased in replisome mutant backgrounds, the large majority of silenced cells still faithfully transmitted the silenced state through cell divisions. Indeed, though *dpb3Δ* and *mcm2-3A* single mutants exhibit asymmetric parental H3-H4 tetramer inheritance (*Yu et al., 2018*; *Petryk et al., 2018*), it is likely that this asymmetry is not complete and some parental H3-H4 tetramers are still stochastically transmitted to each daughter chromatid during DNA replication. Similarly, the *dpb3Δ mcm2-3A* double mutant exhibits residual local inheritance of histone H4 (*Schlissel and Rine, 2019*). We reasoned that, if a daughter chromatid consistently inherits fewer parental H3-H4 tetramers and thereby loses the silenced state more frequently, an additional reduction in the size of a chromatin domain would cause that daughter chromatid to inherit even fewer marked parental H3-H4 tetramers and experience loss-of-silencing events even more frequently. Therefore, if parental H3-H4 tetramers carry epigenetic memory, we would expect loci with fewer nucleosomes to exhibit more loss-of-silencing events in replisome mutant backgrounds. To test this idea, we used the FLAME assay on nucleosome-number mutants in *dpb3Δ* and *dpb3Δ mcm2-3A* strains (*Figure 5A*, *Figure 5—figure supplement 1*). There was no clear correlation between silencing-loss rates and nucleosome number in these sensitized backgrounds (*Figure 5B*). Establishment-of-silencing rates were also not strongly affected, though there was a small increase in the establishment rate with fewer nucleosomes in *dpb3Δ mcm2-3A* (*Figure 5C*). Therefore, even when parental H3-H4 tetramer inheritance was disrupted and the number of parental H3-H4 tetramers available for inheritance at *HMR* was decreased, cells faithfully transmitted epigenetic transcriptional states.

## Discussion

Heterochromatin is frequently characterized by specific histone modifications bound by silencing proteins; these components are critical to mechanisms of silencing and have long been considered as mediators of epigenetic inheritance. A popular model is that modified H3-H4 tetramers are heritable units of epigenetic information that are randomly segregated between daughter chromatids during DNA replication (*Ramachandran and Henikoff, 2015*). Models founded on random segregation of parental H3-H4 tetramers predict that shorter chromatin domains would decrease the heritability of chromatin states in those domains. Contrary to the prediction, we found that shortening the silenced chromatin domain at *HMR* had no significant effects on silencing-loss rate as measured by the CRASH and FLAME assays, even in mutants lacking a component of the silencer-binding complex and in mutants with defective versions of two different regulators of parental H3-H4 tetramer segregation.

### Evidence that H3-H4 tetramers did not carry epigenetic memory

Removal of silencers from heterochromatin via induced recombination demonstrates that silencers are necessary for maintenance of the silenced state. Specifically, induced silencer excision from *HMR* causes rapid loss of silencing in arrested cells (*Cheng and Gartenberg, 2000*). Studies at other loci

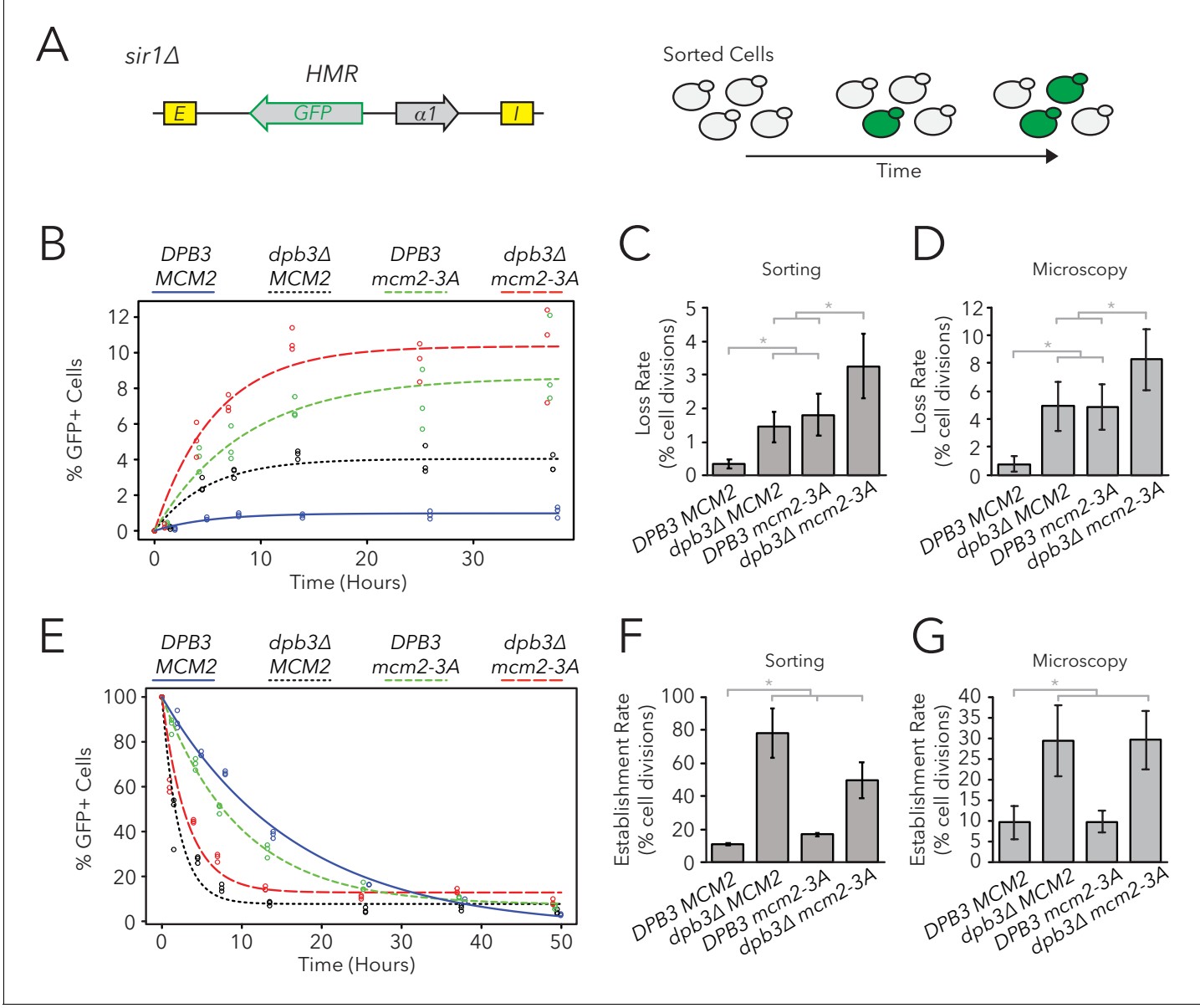

**Figure 4.** Replisome mutants exhibited defects in epigenetic inheritance in the FLAME assay. (A) FACS-based approach to measure switching rates of *HMRα::GFP* in *sir1Δ*. Populations of silenced cells were isolated and allowed to divide; as silencing loss occurred, the percentage of expressed cells in the population increased. The distributions of fluorescence intensity per cell at equilibrium are shown in *Figure 4—figure supplement 1*. (B) For *DPB3 MCM2* (blue) (JRY11471), *dpb3Δ MCM2* (black) (JRY11550), *DPB3 mcm2-3A* (green) (JRY11589), and *dpb3Δ mcm2-3A* (red) (JRY11590), silenced cells were isolated at t = 0 hr, allocated into three separate populations each, and monitored over time. At each time-point, the percentage of expressed cells in each population was determined by flow cytometry (for an example, see *Figure 4—figure supplement 2*). (C) Silencing-loss rates calculated from (B), as explained in Materials and methods. (D) Silencing-loss rates calculated by monitoring dividing cells with time-lapse microscopy (n > 550 cell divisions per genotype). (E) Similar to (B), except expressed cells were sorted and monitored over time. (F) Silencing-establishment rates calculated from (E), as explained in Materials and methods. (G) Silencing-establishment rates calculated by monitoring dividing cells with time-lapse microscopy (n > 100 cell divisions per genotype). GFP expression levels in expressed cells were calculated by flow cytometry and shown in *Figure 4—figure supplement 3*. Error bars represent 95% confidence intervals. Two-tailed t-tests were used in statistical analysis of switching rates by sorting, and Yates chi-square tests were used for microscopy (*p<0.05).

DOI: https://doi.org/10.7554/eLife.51421.016

The following figure supplements are available for figure 4:

**Figure supplement 1.** Replisome mutants exhibited different frequencies of silenced and expressed cells in *sir1Δ*.
DOI: https://doi.org/10.7554/eLife.51421.017

**Figure supplement 2.** Flow cytometry profiles of *sir1Δ dpb3Δ mcm2-3A HMRα::GFP* after FACS sorting.
*Figure 4 continued on next page*

*Figure 4 continued*

DOI: https://doi.org/10.7554/eLife.51421.018

**Figure supplement 3.** *dpb3Δ* exhibited a higher expression level of *HMRα::GFP* in expressed cells.

DOI: https://doi.org/10.7554/eLife.51421.019

in *S. cerevisiae* and *Drosophila* show that removal of silencers permits maintenance of silencing in arrested cells, but causes loss of silencing once the same cells subsequently complete one or two rounds of DNA replication (*Holmes and Broach, 1996*; *Laprell et al., 2017*). Therefore, the presence of modified histones is not sufficient for silencing maintenance or heritability, depending on the example under consideration. Indeed, given that silencers are constantly recruiting Sir proteins to these loci, any role of H3-H4 tetramers in transmission of epigenetic information might be hard to detect.

We considered the possibility that silencer activity masks an underlying contribution of H3-H4 tetramer inheritance to silencing inheritance. However, the weakened silencer activity in *sir1Δ* mutants did not reveal a sensitivity of silencing inheritance to the size of the silenced domain at *HMR*. Importantly, epigenetic states of *HML* and *HMR* in *sir1Δ* are a property of the locus rather than the cell, demonstrating that factors that determine these epigenetic states are inherited locally at *HML* and *HMR* respectively (*Xu et al., 2006*). Similar studies of an epigenetically-inherited heterochromatin state in *Arabidopsis* also demonstrate that the relevant epigenetic information is carried in *cis* (*Berry et al., 2015*). Additionally, epigenetic inheritance of transcriptional states in heterochromatin is commonly accompanied by the ability to switch stochastically between states, a feature that implies the existence of imperfectly heritable epigenetic information. Though modified H3-H4 tetramers could theoretically be *cis*-acting, imperfectly heritable units of information, our evidence to the contrary suggests that other *cis*-acting factors determine the epigenetic state of *HMR* in *sir1Δ*. Given the importance of silencers in inheritance of the silenced chromatin state, one possibility is that the silencer complex self-templates by cooperative oligomerization of silencing factors, and that stochastic changes in epigenetic states reflect the formation or dissolution of such a silencer complex.

## Addressing the possibility that tetramer inheritance is not random

Classic studies of chromatin replication indicate that parental H3-H4 tetramers are randomly segregated between daughter chromatids during DNA replication. For example, chromatin replicated in the presence of cycloheximide, which blocks the synthesis of new histones, produces daughter chromatids with roughly half the number of nucleosomes, and these nucleosomes appear randomly dispersed along both daughter chromatids (*Sogo et al., 1986*; *Cusick et al., 1984*). Though our experiments built on these classic findings, it is also possible that parental H3-H4 tetramers may not be randomly segregated genome wide, or at *HMR* in particular. For example, it was possible that heterochromatin contained factors that facilitated alternating inheritance of tetramers between the leading and lagging strands. In this case, even if H3-H4 tetramers were to act as the sole units of epigenetic information, decreasing chromatin domain size might not affect the rate of silencing loss at *HMR*.

If H3-H4 tetramers carry epigenetic information through DNA replication, mutations that reduce tetramer inheritance would be expected to increase the frequency of silencing loss. Studies describe roles of Dpb3 and Mcm2 in heterochromatic silencing at *HML* (*Yu et al., 2018*; *Gan et al., 2018*), and inheritance of epigenetic states at a synthetic telomere (*Iida and Araki, 2004*; *Foltman et al., 2013*). Using the CRASH and FLAME assays, we found mild but significant increases in *HMR* silencing-loss rates in both *dpb3Δ* and *mcm2-3A* single mutants. Additionally, the *dpb3Δ mcm2-3A* double mutant exhibited higher silencing-loss rates than either of the single mutants. Together, these effects suggested that reduced tetramer inheritance caused mild defects in silencing heritability. Though deacetylated H4K16 is crucial for silencing, other modifications such as H3K56 acetylation also affect silencing (*Hyland et al., 2005*; *Xu et al., 2007*) and reduced inheritance of these modifications may hinder their functions. Considering the variety of histone modifications that parental H3-H4 tetramers can carry through DNA replication, it was striking that cells with moderate or severe

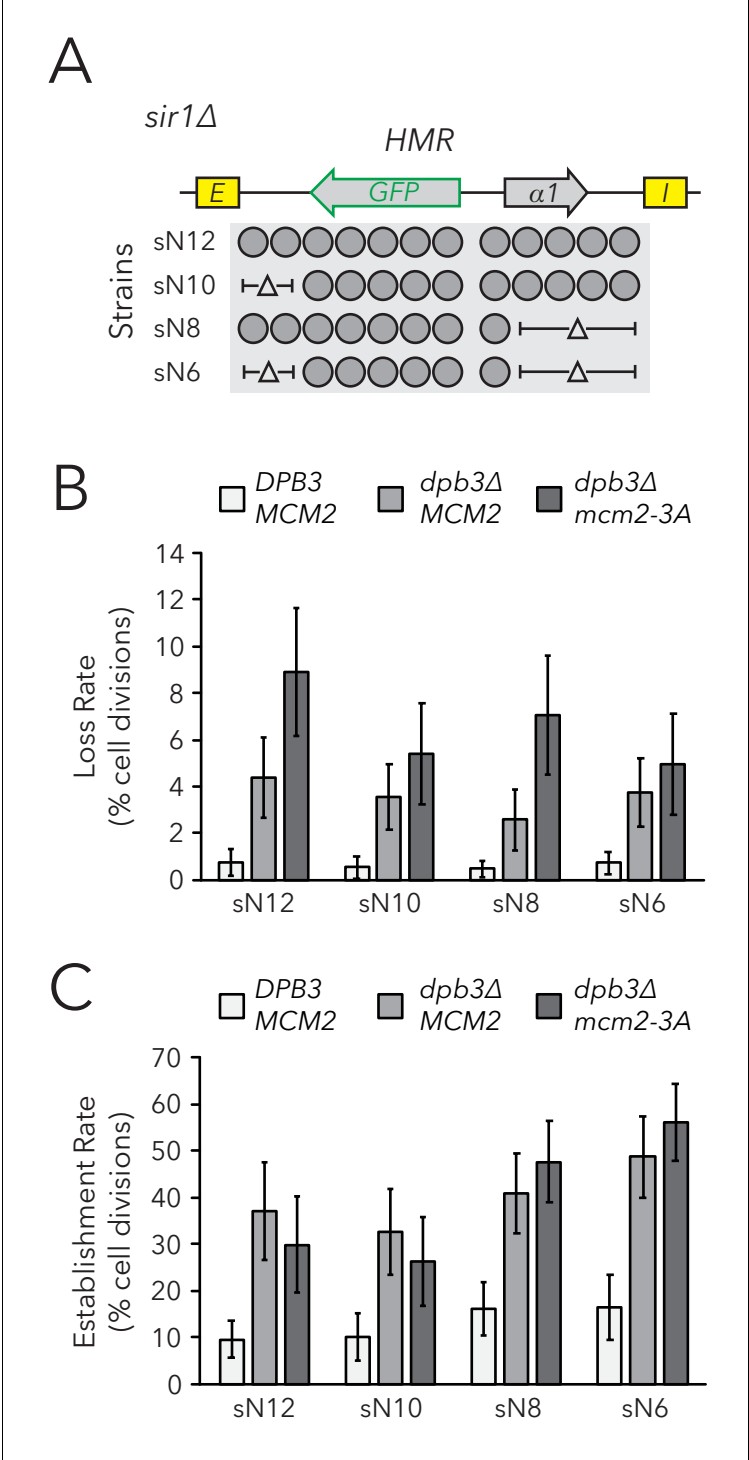

**Figure 5.** Chromatin domain size did not strongly affect epigenetic switching rates in replisome mutant backgrounds. (**A**) Diagram of nucleosomes in *HMRα::GFP*, as seen in *Figure 2B*. As before, combinations of nucleosomal DNA were deleted to change the size of *HMRα::GFP*; the largest allele contained twelve nucleosomes (Strain sN12) (JRY11478) and the smallest allele contained six nucleosomes (Strain sN6) (JRY11547). Frequencies of silenced and expressed cells in these strains were measured by flow cytometry and shown in *Figure 5—figure supplement 1*. (**B**) Loss-of-silencing rates in the FLAME assay. Replisome mutant strains *DPB3 MCM2* (JRY11478) (white), *dpb3Δ MCM2* (JRY11550) (gray), and *dpb3Δ mcm2-3A* (JRY11590) (dark gray) with different numbers of nucleosomes at *HMRα::GFP* were analyzed by time-lapse microscopy (n > 300 cell divisions for each genotype). (**C**) Establishment-of-silencing rates for the same strains as in (**B**), calculated by time-lapse

*Figure 5 continued on next page*

*Figure 5 continued*

microscopy (n > 80 cell divisions per genotype). Loss and establishment rates of *DPB3 MCM2* (JRY11478) are identical to those in **Figure 2D,E** and shown here for convenience. Error bars represent 95% confidence intervals.

DOI: https://doi.org/10.7554/eLife.51421.021

The following figure supplement is available for figure 5:

**Figure supplement 1.** Chromatin domain size of *HMRα::GFP* did not strongly affect the frequencies of different epigenetic states in replisome mutant backgrounds.

DOI: https://doi.org/10.7554/eLife.51421.022

reductions in inheritance of parental H3-H4 tetramers still exhibited efficient inheritance of the silenced state.

Though replisome mutants exhibit defects in parental H3-H4 tetramer inheritance, some tetramers are still transmitted to both daughter chromatids in replisome mutant backgrounds (**Yu et al., 2018**; **Gan et al., 2018**; **Schlissel and Rine, 2019**). Therefore, there are still parental tetramers that are theoretically capable of carrying epigenetic information to both daughter chromatids in the *dpb3Δ*, *mcm2-3A*, and *dpb3Δ mcm2-3A* mutants. Given that all replisome mutants tested showed increased silencing-loss rates, further reduction in the number of parental H3-H4 tetramers available for transmission to daughter chromatids should cause even higher rates of silencing loss. However, we saw no significant effects of *HMR* size on the silencing-loss rate in replisome mutant backgrounds. Therefore, cells with both reduced parental H3-H4 tetramer inheritance and a reduction in the number of tetramers available for inheritance at *HMR* exhibited a surprisingly robust ability to transmit the silenced state. These data strongly suggested that inheritance of parental H3-H4 tetramers has little or no impact on epigenetic inheritance of the silenced state of *HMR*.

## Epigenetic inheritance of the expressed state

The expressed state of *HMR* in *sir1Δ* cells is formally an epigenetic state: it is heritable through cell divisions and can stochastically switch to the silenced state. One possibility is that the expressed state of *HMR* depends on the existence of heritable information, similarly to the silenced state. Histone modifications associated with active transcription can be transmitted through DNA replication (**Alabert et al., 2015**; **Reverón-Gómez et al., 2018**) and multiple transcription factors can bind to the histone modifications they generate (**Jacobson et al., 2000**; **Owen et al., 2000**). Therefore, histone modifications may form positive feedback loops with both silencing machinery and transcription factors. Indeed, a model that incorporates these positive feedback loops and parental H3-H4 tetramer inheritance generates robust bistable chromatin states (**Dodd et al., 2007**). This model also predicts that random segregation of parental H3-H4 tetramers would lead to loss-of-chromatin-state events, and that decreasing chromatin domain size would also decrease the heritability of both the expressed and silenced states. However, we found that shorter versions of *HMR* did not strongly affect inheritance of the expressed state of *HMR*.

Alternatively, if parental H3-H4 tetramers carry memory of the expressed state, mutations that disrupt parental H3-H4 tetramer inheritance would be expected to increase the rate of silencing establishment. Curiously, *dpb3Δ* exhibited a ~3-fold increase in the rate of silencing establishment and *mcm2-3A* had no observable effect (see **Figure 4G**). These data may suggest that parental tetramer inheritance facilitates heritability of the expressed state, though such an explanation could not account for the *mcm2-3A* phenotype. Alternatively, these data may suggest that inheritance of the expressed state is influenced by a function of Dpb3 that is separate from its role in tetramer inheritance. It is also important to note that *dpb3Δ* but not *mcm2-3A* led to elevated levels of GFP expression when *HMRα::GFP* was fully expressed. This finding is paradoxical, as one would expect elevated transcription to inhibit silencing establishment, rather than facilitate it. However, recruitment of the transcriptional activator Ppr1 to *HMR* causes both increased transcription in expressed cells and an increased establishment rate in *sir1Δ* (**Xu et al., 2006**).

Together, our results suggested that the fidelity of H3-H4 tetramer inheritance has minimal consequences for heritability of the silenced state and may affect heritability of the expressed state in some contexts. These findings raised doubts regarding the model in which histones are significant carriers of epigenetic memory in *S. cerevisiae*. As such, future studies that continue to examine

histone-based memory models will be complemented by studies on other possible mechanisms of transcriptional memory.

## Materials and methods

### Yeast strains

The strains and oligonucleotides used in this study are listed in *Supplementary files 1* and *2*, respectively. All strains were derived from the W303 background. CRASH assay strains, which contained *HMRα, hmrα2Δ::cre, ura3Δ::loxP::yEmRFP:tCYC1:KanMX:loxP:yEGFP:tADH1* or *hmlα2Δ::cre, ura3Δ:: loxP::yEmRFP:tCYC1:HygMX:loxP:yEGFP:tADH1* were generated as described previously (*Dodson and Rine, 2015*). FLAME assay strains were generated with the following approach. To generate *hmlα2Δ::yEmRFP*, a *K. lactis URA3* swap was performed to replace the *α2* coding sequence with *yEmRFP* coding sequence. The *hmlα2Δ::yEmRFP* fwd/rev primers were used for integration of *yEmRFP* in the final step. To generate *HMRα, hmrα2Δ::yEGFP*, a fragment spanning a portion of *hmlα2Δ::yEGFP* was amplified using *hmlα2Δ::yEGFP* fwd/rev primers and swapped into *HMRa*.

To delete DNA corresponding to nucleosomes at *HMRα* and *HMLα*, CRISPR/Cas9 was employed as previously described (*Lee et al., 2015*). Each deletion or repair fwd/rev primer set contained two partially overlapping primers that were amplified by PCR prior to use. The *HMR-E*-proximal sgRNA was used to induce Cas9 cutting between the *HMR-E* silencer and *cre*, and N14 to N12 deletion fwd/rev was used to delete DNA corresponding to two nucleosomes in this region. This sgRNA and oligo set was also used to convert sN12 to sN10 in the FLAME strain background. The *HMR-I*-proximal sgRNA, which cuts between the *HMR-I* silencer and *cre*, was used with N14 to N10 deletion fwd/rev (to convert N14 to N10, and sN12 to sN8) or with N14 to N9 deletion fwd/rev (to convert N14 to N9). For *HMLα*, the *HML-E*-proximal sgRNA was used to induce Cas9 cutting between the HML-I silencer and *cre*, and used with N22 to N19a deletion fwd/rev (to convert N22 to N19a) or N22 to N16a deletion fwd/rev (to convert N22 to N16a). The *HML-I*-proximal sgRNA was used to induce Cas9 cutting between the *HML-E* silencer and *cre*, and was used with N22 to N19b deletion fwd/rev (to convert N22 to N19b) or N22 to N16c deletion fwd/rev (to convert N22 to N16c) or N22 to N13c deletion fwd/rev (to convert N22 to N13c). Deletions were confirmed by junction primers and sequencing. To generate mutants with combinations of nucleosome set deletions, CRISPR/Cas9 technology was applied (as described above) to strains with one nucleosome set deletion already made.

To generate *dpb3Δ*, the DPB3 sgRNA was used with Cas9 to cut within *DPB3* and *DPB3* deletion fwd/rev was used to delete the coding sequence. To generate *mcm2-3A*, the *MCM2* sgRNA was used with Cas9 to cut 244 bp into the *MCM2* coding sequence and *mcm2-3A* repair fwd/rev was used to generate the appropriate point mutations (Y79A Y82A Y91A). Mutations were confirmed by sequencing.

### Colony growth and imaging

To generate colonies for analysis by the CRASH assay, RFP-expressing cells were diluted and plated at a density of ~10 cells/plate (CSM-Trp (Sunrise Science Products, San Diego, CA), 1% agar). After 5 days of growth, colonies were imaged using a Leica M205 FA fluorescence stereomicroscope (Leica Camera AG, Wetzlar, Germany) equipped with a Leica DFC3000G CCD camera, a Leica PLANAPO 0.63x objective, ET RFP filter (Leica 10450224), ET GFP filter (Leica 10447408), and Leica Application Suite X (LAS X) imaging software. At least ten colonies were imaged per genotype.

### Live-cell imaging

Cells were grown to saturation in CSM (Sunrise Science Products) at 30°C overnight. These cells were then back-diluted in 5 ml CSM and grown to mid-log phase over 6 hr. 500 μl was transferred to a microfuge tube and sonicated at 20% for 15 s (Branson Ultrasonics Digital Sonifier 100-132-888R with Sonicator Tip 101-135-066R) (Branson Ultrasonics, Fremont, CA) to break up clumps of cells. 5 μl of sonicated cells were spotted onto a CSM plate (1% agar) and allowed to soak into the agar. When dry, a sterile spatula was used to cut a 1 cm ×1 cm agar square surrounding the cell patch. The square was lifted out of the plate, inverted, and placed in a 35 mm glass bottom dish (Thermo Scientific 150682) (Thermo Fisher Scientific, Waltham, MA). Cells were imaged using a Zeiss

Z1 inverted fluorescence microscope with a Prime 95B sCMOS camera (Teledyne Photometrics, Tucson, AZ), Plan-Apochromat 63x/1.40 oil immersion objective (Zeiss, Oberkochen, Germany), filters, MS-2000 XYZ automated stage (Applied Scientific Instrumentation, Inc, Eugene, OR), and Micro-Manager imaging software (Open Imaging, San Fransisco, CA). Given that cells were pressed between the agar and glass, the cells were all in the same focal plane and Z-stacks were not used.

For time-lapse microscopy (i.e. *Figure 2D*), samples were kept at 30℃ and humidified with a P-Set 2000 Heated Incubation Insert (PeCon, Erbach, Germany). Time-lapse experiments involved brightfield and fluorescence imaging of 16 different fields per sample, and images were taken every 10 min for 10 hr. Subsequent analysis of cell divisions was done in ImageJ (NIH, Bethesda, MD). To measure epigenetic switching rates in the FLAME assay, cell divisions and switching events were manually counted and the counter was blind to the genotype (single-blind study). This counting was performed only on cells that could be clearly distinguished from each other. If a mother and daughter cell pair switched simultaneously, we counted this as one switching event that probably appeared as two events due to the lag time in yEGFP expression or degradation.

## Flow cytometry

To measure fluorescence intensities per cell in the CRASH and FLAME assays, a BD LSR Fortessa cell analyzer (BD Biosciences, San Jose, CA) with a FITC filter (for GFP) and a PE-TexasRed filter (for RFP) was used. Subsequent analysis was performed with FlowJo software.

For quantification of silencing-loss rates in the CRASH assay (*Figure 1F*; *Figure 3B*), cells were first streaked out to form single colonies. Six colonies per genotype were added to CSM-Trp media (Sunrise Science Products) in a 96-well plate (Corning CLS3788) (Corning Inc, Corning, NY) and grown to saturation overnight in an incubating microplate shaker (VWR 12620–930) (VWR International, Radnor, PA) at 30℃. These samples were then back-diluted and grown to mid-log phase over 6 hr. GFP and RFP expression were then analyzed by flow cytometry (n > 4000 cells per sample). Distinct populations of RFP+ GFP- (which had not lost silencing), RFP+ GFP+ (which had recently lost silencing), and RFP- GFP+ (which had lost silencing less recently) were observed. The apparent silencing-loss rate was calculated as the number of RFP+ GFP+ cells divided by the number of RFP+ GFP+ cells and RFP+ GFP- cells. Measurements from independent cultures were considered as biological replicates.

For calculating the frequency of silenced and expressed cells at equilibrium in the FLAME assay, cells were first streaked out to generate single colonies. Three colonies per genotype were added to CSM media in a 96-well plate and grown to saturation overnight. These samples were then serially back-diluted in CSM media in 96-well plates and grown at 30℃. After twelve hours, the serial dilutions had a range of cell densities; the dilution that was closest to ~1 O.D. was again back-diluted in CSM media and grown at 30℃ for another 12 hr. At this point, wells close to ~1 O.D. contained cells that had been growing at log-phase for approximately 24 hr. These cells were analyzed by flow cytometry. Because three populations were analyzed per genotype, the most representative profiles of silenced and expressed cells were used for figures. We considered these populations as biological replicates.

To calculate GFP expression levels in expressed cells in the FLAME assay, cells were streaked out for single colonies and three colonies per genotype were grown overnight in CSM + 5 mM Nicotinamide (NAM) (Sigma-Aldrich, St. Louis, MO). These samples were then back-diluted in CSM + 5 mM NAM and grown at 30℃ for 12 hr. Samples at ~1 O.D. were analyzed by flow cytometry. For *Figure 2—figure supplement 4*, the most representative profiles of the three profiles generated per strain were shown. For *Figure 4—figure supplement 3*, the geometric mean intensity of GFP per cell (excluding cells that formed a smaller, artifactual peak at a lower GFP intensity) was calculated for each population using FlowJo software. Independent cultures were considered as biological replicates.

FACS was utilized in the FLAME assay to calculate switching rates between epigenetic states in *Figure 4*. To perform this experiment, cells from each genotype were serially diluted in CSM media and grown at 30℃. After 12 hr, dilutions closest to ~1 O.D. were sorted into GFP- and GFP+ populations using a BD FACSAria Fusion cell sorter (BD Biosciences) equipped with a FITC filter for GFP. Gates were calibrated from *SIR⁺* (JRY11474) and *sir4Δ* (JRY11496) cells. For each sample, 150,000 GFP- cells were sorted into one tube and 30,000 GFP+ cells were sorted into another. Each sorted population was divided evenly into three populations and grown in CSM in a 96-well plate at 30℃.

Serial back-dilutions were used to maintain constant log-phase growth over two days. Time-points were taken by removing a fraction of cells from each population and fixing them in a 4% paraformaldehyde solution (4% Paraformaldehyde, 3.4% Sucrose) for 15 min at room temperature. Fixed cells were resuspended in GFP fix buffer (100 mM KPO$_4$ pH 7.4, 1.2 M Sorbitol) and kept at 4°C. Once the experiment was complete, fixed cells from different time-points were analyzed by flow cytometry (n > 500 cells per sample) and FlowJo software. The percent of GFP+ cells for each sample over time is shown in *Figure 4B and E*. Because the initial sorting event required ~20 min per sample, the time of initial sorting (t = 0 hr) was different between samples; this made the time points between samples slightly staggered as seen in *Figure 4B and E*. Because cells were divided into subpopulations after the initial sorting, these subpopulations were considered as technical replicates.

## Switching rate calculation from cell sorting

The following equations were used to model the dynamics of switching rates between epigenetic states in *sir1Δ*. We considered the balance of GFP+ and GFP- cells over time, and assumed that the birth and death rates of the two populations are similar. Combining the balances and introducing the ratio variable $x$, we can derive the following equation that describes how a population of GFP+ cells and GFP- cells would move towards equilibrium over time:

$$\left(\frac{1}{k_{ON}+k_{OFF}}\right)\frac{dx_{ON}}{dt}+x_{ON}=\frac{k_{ON}}{k_{ON}+k_{OFF}}$$

$k_{ON}$ is the loss rate per hour, $k_{OFF}$ is the establishment rate per hour, $x_{ON}$ is the fraction of GFP+ cells at a given time, and $t$ is time. Solving the differential equation for $x_{ON}$ yields:

$$x_{ON}=\frac{k_{ON}}{k_{ON}+k_{OFF}}\left(1-e^{-\frac{t}{k_{ON}+k_{OFF}}}\right) \quad \text{or} \quad x_{ON}=\frac{k_{ON}}{k_{ON}+k_{OFF}}\left(1-e^{-\frac{t}{k_{ON}+k_{OFF}}}\right)+e^{-\frac{t}{k_{ON}+k_{OFF}}}$$
$$\text{if } x_{ON}=0 \text{ at t}=0 \qquad\qquad\qquad\qquad \text{if } x_{ON}=1 \text{ at t}=0$$

Therefore, the following equations were used to model switching rates between epigenetic states from data in *Figure 4B and E*.

Sorting silenced cells (*Figure 4B*):

$$x_{ON}=\frac{k_{ON}}{k_{ON}+k_{Off}}(1-e^{-\frac{t}{k_{ON}+k_{OFF}}})$$

Sorting expressed cells (*Figure 4E*):

$$x_{ON}=\frac{k_{ON}}{k_{ON}+k_{Off}}(1-e^{-\frac{t}{k_{ON}+k_{OFF}}})+e^{-\frac{t}{k_{ON}+k_{OFF}}}$$

The `nls()` function in R was used to provide a nonlinear least squares estimate of the unknown variables $k_{ON}$ and $k_{OFF}$ for each genotype, and 95% confidence intervals for estimates. With this approach, each genotype had an estimated $k_{ON}$ and $k_{OFF}$ from sorting silenced cells and an estimated $k_{ON}$ and $k_{OFF}$ from sorting expressed cells. Since sorting silenced cells subsequently allowed for observation of more loss-of-silencing events, the $k_{ON}$ rates from those data were considered more accurate and used in *Figure 4C*. Similarly, the $k_{OFF}$ rates calculated from sorting expressed cells were used in *Figure 4F*.

Because each population of sorted cells was evenly divided into three subpopulations, each genotype has three calculated values for the percent of GFP+ cells at each given time point after sorting. The nonlinear least squares estimate was made by drawing a best fit line through all data points for a given genotype, effectively combining the values of all subpopulations. The quality of the fit was calculated using the `confint2()` function and represented as 95% confidence intervals for $k_{ON}$ values in *Figure 4C* and $k_{OFF}$ values in *Figure 4F*. An alternative approach involved drawing a best fit line for each individual subpopulation to give three $k_{ON}$ values and three $k_{OFF}$ values for each genotype and averaging these values to get a single $k_{ON}$ value and $k_{OFF}$ value for each genotype, with error bars representing a standard deviation. Though we also performed this latter analysis method, we favor the former analysis method because it incorporates how well the data fit the nonlinear least squares estimate. Notably, both analysis methods gave similar $k_{ON}$ and $k_{OFF}$ values.

The generation time of *DPB3 MCM2* (JRY11471) was 1.96 hours in CSM media at 30°C. To convert $k_{ON}$ and $k_{OFF}$ as rates per hour to rates per generation, we multiplied these variables by the generation time. Similar generation times were observed for all replisome mutants.

## MNase-Seq

Cells were grown to saturation overnight in 5 mL CSM at 30°C. The following day, these cells were back-diluted to ~0.1 O.D. in 50 ml CSM and grown at 30°C for 5 hr. Cells were then centrifuged and washed twice in 500 µl SKC buffer (1.2 M Sorbitol, 100 mM $KH_2PO_4$, 0.5 mM $CaCl_2$, 7 mM β-mercaptoethanol) and then resuspended in 100 µl SKC buffer. Cells were incubated at 37°C for 15 min, then 30 µl of 1 mg/mL Zymolyase-100T (MP Biomedicals, LLC, Solon, OH) was added for a final concentration of 0.23 mg/ml Zymolyase-100T and incubated at 37°C for 15 min. All subsequent steps were performed on ice and subsequent centrifugations performed with an accuSpin Micro 17R (Fischer Scientific, Hampton, NH). Once spheroplasting was complete, cells were spun at 3 k RPM for 3 min at 4°C. Cells were washed twice in 500 µl SPC buffer (1 M Sorbitol, 20 mM PIPES pH 6.3, 0.1 mM $CaCl_2$, with Roche cOmplete protease inhibitors (Sigma)) and spun at 2 k RPM for 3 min at 4°C between washes. Cells were resuspended in 250 µl SPC buffer, and this solution was gently mixed with 250 µl freshly prepared Ficoll buffer (9% Ficoll, 20 mM PIPES pH 6.3, 0.5 mM $CaCl_2$) to lyse the cell membranes.

Nuclei were then pelleted by centrifugation at 10 k RPM for 20 min at 4°C. Nuclei were washed twice in 500 µl SPC and spun at 8 k RPM for 3 min at 4°C between washes. Washed nuclei were subsequently resuspended in 250 µl SPC and $CaCl_2$ was added to a final concentration of 2 mM $CaCl_2$. Nuclei were incubated for 5 min at 37°C, then 20 units of Worthington MNase was added (Worthington Biochemical Corporation, Lakewood, NJ). Nuclei were incubated for 15 min at 37°C. MNase activity was quenched by addition of EDTA to a final concentration of 10 mM EDTA. Nuclei were centrifuged at 3.7 k RPM for 5 min at 4°C. The nucleosome-containing supernatant was subsequently removed and DNA and RNA were purified using a Qiagen spin column. RNase A (Sigma) was added to a final concentration of 1 mg/ml RNase A and incubated for 2 hr at 37°C. DNA was then purified using a Qiagen spin column. MNase libraries were constructed with NEBnextUltra II library preparation kit (New England Biolabs, Ipswich, MA) and sequenced on an Illumina HiSeq4000 (Illumina, San Diego, CA) as 100 bp paired-end reads.

Reads were mapped to the *Saccharomyces cerevisiae* S288C genome (GenBank accession number GCA_000146045.2) using Bowtie2 (*Langmead and Salzberg, 2012*). Mapped reads between 140 bp and 180 bp in length were used in all further analysis to ensure mononucleosome resolution. The midpoint for each read was calculated and midpoints were stacked in a histogram. Finally, a 25 bp rolling mean was used to smooth out the resulting nucleosome peaks. All sequences and processed data files have been deposited in the NCBI Gene Expression Omnibus archive under accession number GSE136897.

## Acknowledgements

We thank Anne Dodson for providing strains and genetic tools that were integral to this study. We also thank the UC Berkeley Flow Cytometry facility, especially Hector Nolla, for use of flow cytometers. This work used the Vincent J Coates Genomics Sequencing Laboratory at UC Berkeley, supported by NIH S10 OD018174 Instrumentation Grant. We are grateful to Marc Fouet for help with microscopy and modeling switching rates. We thank Albert Serra-Cardona for discussions on replisome factors and histone inheritance patterns. Finally, we thank Rine laboratory members for thoughtful discussions, especially Gavin Schlissel, Davis Goodnight, Molly Brothers, Ryan Janke, Eliana Bondra, and Delaney Farris.

## Additional information

### Funding

| Funder | Grant reference number | Author |
| --- | --- | --- |
| National Institutes of Health | GM 031105 | Jasper Rine |

| National Institutes of Health | GM 120374 | Jasper Rine |
| National Science Foundation | DGE 1752814 | Daniel S Saxton |

The funders had no role in study design, data collection and interpretation, or the decision to submit the work for publication.

### Author contributions
Daniel S Saxton, Conceptualization, Resources, Data curation, Software, Formal analysis, Validation, Investigation, Visualization, Methodology, Writing—original draft, Project administration, Writing—review and editing; Jasper Rine, Conceptualization, Resources, Supervision, Funding acquisition, Writing—review and editing

### Author ORCIDs
Daniel S Saxton (ID) https://orcid.org/0000-0001-7152-7780
Jasper Rine (ID) https://orcid.org/0000-0003-2297-9814

### Decision letter and Author response
Decision letter https://doi.org/10.7554/eLife.51421.029
Author response https://doi.org/10.7554/eLife.51421.030

## Additional files

### Supplementary files
• Supplementary file 1. Table of yeast strains used in this study.
DOI: https://doi.org/10.7554/eLife.51421.023
• Supplementary file 2. Table of oligonucleotides used in this study.
DOI: https://doi.org/10.7554/eLife.51421.024
• Transparent reporting form  DOI: https://doi.org/10.7554/eLife.51421.025

### Data availability
Sequencing data have been deposited in GEO under accession code GSE136897.

The following dataset was generated:

| Author(s) | Year | Dataset title | Dataset URL | Database and Identifier |
|---|---|---|---|---|
| Saxton DS, Rine J | 2019 | Nucleosome profiles in strains with different numbers of nucleosomes at HML and HMR | https://www.ncbi.nlm.nih.gov/geo/query/acc.cgi?acc=GSE136897 | NCBI Gene Expression Omnibus, GSE136897 |

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
