## [Decision Letter]

Thank you for submitting your article "Epigenetic memory independent of symmetric histone inheritance" for consideration by *eLife*. Your article has been reviewed by 3 peer reviewers, including Tim Formosa as the Reviewing Editor and Reviewer #1, and the evaluation has been overseen by Kevin Struhl as the Senior Editor. The following individuals involved in review of your submission have agreed to reveal their identity: Marc Gartenberg (Reviewer #2).

The reviewers have discussed the reviews with one another, and the Reviewing Editor has drafted this decision to help you prepare a revised submission.

Summary:

In this manuscript, Saxton and Rine test the idea that epigenetic inheritance of silencing in *Saccharomyces cerevisiae* is mediated by segregation of histone H3-H4 tetramers. Specifically, they test the prediction that if inheritance of a single parental H3-H4 tetramer in a silenced region is adequate to regenerate the silent state in the daughter, loss of silencing will occur more frequently if the number of nucleosomes in the silenced array is reduced as this will increase the frequency of inheriting no parental H3-H4 tetramers. The results do not support this model, challenging the idea that modifications of H3-H4 provide the core information for epigenetic inheritance of silencing. Potential confounding effects of Sir1 are examined and discarded, and mutations that affect the segregation bias of H3-H4 to leading and lagging strands during replication are also tested. The latter show some effects on silencing, as expected from previous work, but do not provide support for the model that H3-H4 tetramers carry the information to determine expression state in daughters. The key impacts of the work are therefore failure to find support for a commonly discussed feature of epigenetic inheritance models, and the development or improvement of important tools for examining inheritance of silencing in the CRASH and FLAME assays. The presentation is clear, and the data presented are of high quality and are interpreted appropriately. While no new experiments are proposed by the reviewers, several issues were encountered regarding the presentation, as outlined below.

Essential revisions:

1) The precise model being tested and the logic behind the prediction that reducing the number of nucleosomes in the silenced region would lead to increased loss of silencing was not immediately apparent to all reviewers. The authors should highlight the discussion in the Ramachandran and Henikoff reference more prominently and might consider adding a panel to the first figure to illustrate the model. An array of 14 nucleosomes with random distribution predicts only one in 16,000 cells will fail to inherit any parental H3-H4 tetramers, while this should occur once in 128 cells with 7 nucleosomes, so a panel showing these outcomes might make the model more accessible. This raises another issue the authors should address explicitly: the derepression rate with 14 nucleosomes is about 0.1% but the model above predicts it should be 0.006%. Some comment on a mechanism that might account for this 20-fold discrepancy between expectation and the observed values in Figure 1C and Figure 2C should be provided.

A related issue is found in subsection “Nucleosome number did not determine the rate of silencing loss”. The expected loss rates for N14 to N7 span from 0.006% and 0.8%, or 100-fold. This point is made in Figure 1C and Figure 2C but not explicitly mentioned in the text. Instead, the text juxtaposes the expectation for N7 (~1%) with the actual value for wt (0.1%). This unintended comparison, as written, may mislead the casual reader.

2) While DPB3 and MCM2 have been shown to control histone segregation near replication origins, it has not been shown that they act similarly in heterochromatic regions. To do so at *HMR* might be feasible, but would be well beyond the scope of this study. However, the authors must acknowledge that this assumption has been made in interpreting their results.

3) The authors need to explain how the nucleosome number in the FLAME constructs is determined. For the CRASH assay, the number of nucleosomes in the parental and key derivative strains is validated by MNase-seq, but similar data are not provided here for the FLAME constructs. It may not be necessary to provide MNase-seq data, but the authors should explain where the numbers come from and their reasoning for not presenting this validation.

4) The authors acknowledge that the biases incurred by DPB3 and MCM2 are not likely to be absolute. It might be interesting to plug some numbers based on estimates of Yu et al., 2018. For example, if *dpb3* causes a two-fold bias at each of the 14 nucleosomes of *HMR*, there would be a 50-fold greater probability of derepression. This clearly isn't seen in the experiments, so it might be helpful to comment on this.

---

## [Author Response]

Essential revisions:1) The precise model being tested and the logic behind the prediction that reducing the number of nucleosomes in the silenced region would lead to increased loss of silencing was not immediately apparent to all reviewers. The authors should highlight the discussion in the Ramachandran and Henikoff reference more prominently and might consider adding a panel to the first figure to illustrate the model. An array of 14 nucleosomes with random distribution predicts only one in 16,000 cells will fail to inherit any parental H3-H4tetramers, while this should occur once in 128 cells with 7 nucleosomes, so a panel showing these outcomes might make the model more accessible. This raises another issue the authors should address explicitly: the derepression rate with 14 nucleosomes is about 0.1% but the model above predicts it should be 0.006%. Some comment on a mechanism that might account for this 20-fold discrepancy between expectation and the observed values in Figure 1C and Figure 2C should be provided.A related minor issue is found in subsection “Nucleosome number did not determine the rate of silencing loss”. The expected loss rates for N14 to N7 span from 0.006% and 0.8%, or 100-fold. This point is made in Figure 1C and Figure 2C but not explicitly mentioned in the text. Instead, the text juxtaposes the expectation for N7 (~1%) with the actual value for wt (0.1%). This unintended comparison, as written, may mislead the casual reader.

We expanded the text to accommodate a clearer explanation of the random segregation model and provided a new panel in Figure 1 (Figure 1C) to help readers visually understand this concept. The discrepancy between expected (0.006%) and observed (0.1%) silencing loss rates is also now addressed in the text as due to the existence of other processes besides histone inheritance that potentially destabilize silencing. These changes are present in the Results section, as follows:

“At the limit of models by which nucleosomes transmit memory of transcriptional states, inheritance of a single parental H3-H4 tetramer to a daughter chromatid would be sufficient to template the silenced state. […] Surprisingly, decreasing nucleosome number at *HMRα::cre* led to a slight decrease in silencing loss as measured by sector frequency (Figure 1E).”

2) While DPB3 and MCM2 have been shown to control histone segregation near replication origins, it has not been shown that they act similarly in heterochromatic regions. To do so at HMR might be feasible, but would be well beyond the scope of this study. However, the authors must acknowledge that this assumption has been made in interpreting their results.

This assumption is now explicitly addressed in the Results section, as follows:

“Additionally, since previous studies did not specifically test the effects of Dpb3 and Mcm2 on histone inheritance within heterochromatin, any interpretations of silencing defects operated under the assumption that these replisome components act similarly between heterochromatin and euchromatin.”

We note that although we are making an assumption that Dpb3 and Mcm2 act in heterochromatin analogously to the way they act in euchromatin, the mutant phenotypes do suggest that this assumption is not too much of a stretch.

3) The authors need to explain how the nucleosome number in the FLAME constructs is determined. For the CRASH assay, the number of nucleosomes in the parental and key derivative strains is validated by MNase-seq, but similar data are not provided here for the FLAME constructs. It may not be necessary to provide MNase-seq data, but the authors should explain where the numbers come from and their reasoning for not presenting this validation.

We have these data, and have now added Figure 2—figure supplement 3 to show nucleosome positions of *HMRα::GFP* in the FLAME assay. We have amended the legend to Figure 2 to state that MNase-seq defined the positions of nucleosomes and added a reference to the new supplemental figure.

4) The authors acknowledge that the biases incurred by DPB3 and MCM2 are not likely to be absolute. It might be interesting to plug some numbers based on estimates of Yu et al., 2018. For example, if dpb3 causes a two-fold bias at each of the 14 nucleosomes of HMR, there would be a 50-fold greater probability of derepression. This clearly isn't seen in the experiments, so it might be helpful to comment on this.

We agree that building predictions based on these papers would be very useful. However, a limitation of the Yu and Gan papers on symmetry of histone inheritance in *S. cerevisiae* is that the frequency at which origin-proximal DNA is replicated in one direction versus another is not tested. Though these papers picked origins that fire efficiently, other adjacent origins can outcompete them at some frequency, which makes the exact directionality of replication an unknown variable and complicates downstream calculations of symmetry of histone inheritance. This does not change the interpretations that are drawn from these two elegant papers, but does make us cautious about using their numbers to make precise predictions about silencing dynamics.

In contrast, the Petryk et. al paper from the Groth lab does incorporate the Replication Fork Directionality (RFD), making their measurements more precise and appropriate for building predictions. However, since this study was performed in mammalian cells, we hesitated to use their estimates of symmetry of histone inheritance.